# Empirical Evaluation and Prediction of Protein Requirements for Maintenance and Growth of 18–24 Months Old Thai Swamp Buffaloes

**DOI:** 10.3390/ani11051405

**Published:** 2021-05-14

**Authors:** Siwaporn Paengkoum, Pattaraporn Tatsapong, Nittaya Taethaisong, Thongpea Sorasak, Rayudika Aprilia Patindra Purba, Pramote Paengkoum

**Affiliations:** 1Program in Agriculture, Faculty of Science and Technology, Nakhon Ratchasima Rajabhat University, Muang, Nakhon Ratchasima 30000, Thailand; 2Department of Agricultural Science, Faculty of Agriculture, Natural Resources and Environment, Naresuan University, Phitsanulok 65000, Thailand; pattarapornt@nu.ac.th; 3School of Animal Technology and Innovation, Institute of Agricultural Technology, Suranaree University of Technology, Muang, Nakhon Ratchasima 30000, Thailand; ISZY.Nittaya@gmail.com (N.T.); sorasak.t@sut.ac.th (T.S.); rayudikaapp.007@gmail.com (R.A.P.P.); pramote@sut.ac.th (P.P.)

**Keywords:** feeding trial, growth, maintenance, nutrient evaluation, protein utilization, Thai swamp buffalo

## Abstract

**Simple Summary:**

Thai swamp buffalo is a domesticated swamp buffalo (*Bubalus bubalis*) which has a functional significance for the livestock production system and for the economic benefit of Thailand. For instance, meat supply derived from Thai swamp buffalo is a secondary consideration in recently years. Therefore, there is mounting interest concerning the regulation of the nutrient requirements of Thai swamp buffalo to optimize their production. However, no systematic report is available on the nutrient requirements of the growing Thai swamp buffalo. This study investigates and predicts protein requirement systems that can provide an abundant energy intake and can be included in the 18–24 months old Thai swamp buffalo’s diet, as well as supply the recommended amount of net nitrogen or crude protein requirement to optimize their growth and maintenance. Protein requirement, nutrient utilization, and microorganism profile are included to corroborate the influences mentioned.

**Abstract:**

In some geographical areas and in certain breeding situations, the interpretation of increased gain in the bovine is difficult to investigate. Due to their inherent genetic variations, their energy and protein needs vary as a function of inherent genetic differences, making these requirements difficult to accurately assess in bull species, e.g., Thai swamp buffalo. The study aimed at investigating and predicting protein requirement systems, by the provision of an abundant energy intake of 2.2 Mcal/kg DM for the maintenance and growth of Thai swamp buffaloes using a comparative prolonged feeding trial for 90 days. Sixteen bull Thai swamp buffaloes at the initial (Age: 18–24 months; BW: 233 ± 25.0 kg) were assigned into four treatment groups, four buffaloes each, fed 5.4, 6.6, 8.5, and 10.5% DM crude protein (CP). CP intake, BW, and physiological fluid were determined. The net CP requirements for maintenance and growth of Thai swamp buffaloes were 5.41 g CP/kg W^0.75^ and 0.46 g CP/g average daily gain (ADG), respectively. Our results indicated that CP requirement increases when the BW increases. An increased dietary CP resulted in increased amounts of blood urine nitrogen (N), N absorption, total volatile fatty acid, urinary purine derivative, and the microbial N. Notably, the net CP requirement for growth of Thai swamp buffalo was higher than it reported in NRC, but the maintenance was lower.

## 1. Introduction

A domesticated swamp buffalo (*Bubalus bubalis*) is one of economic-based strategies in tropical livestock production to supply important animal resources [1]. The domestic swamp buffalo plays a crucial role, which extends beyond its supply of primarily draught power, hides, social value, and credit; with its contribution as a meat supply due to stabilized population numbers being a secondary consideration in recent years in Asian countries, including Thailand [2]. Molecular and morphological evidence suggests that swamp buffalo populations have strong phenotypic uniformity and geographic genetic distinction and they have been recorded as dispersing through south-east Asia [2]. As they have a high meat production potential, multiple purposes as agricultural tools, and are well-adapted to the hot-humid tropical climate conditions, swamp buffalo are commonly domesticated in several areas in Thailand. Nevertheless, unsuitable feeding system have regularly been noted as the main constraint in meat production systems involving small ruminant and cattle throughout south-east Asia [3]. In dairy, for instance, the nutrient metabolism of water buffalo, either river buffalo or swamp buffalo (Mediterranean) has been studied for many years and the results are very detailed for growing heifers in large farms and for specialized milk production [4,5]. However, no bibliographical notes are reported for: (1) metabolic status and ovarian function in buffalo heifers fed a low energy or high energy; (2) nitrogen and phosphorus utilization and excretion in dairy buffalo intensive breeding; and (3) protein nutrition and nitrogen balance in buffalo cows. Accordingly, in some geographical areas and in certain breeding situations, it is difficult to interpret data of increased gain as a result of greater nutrient metabolism for the bovine animals. Since the nutrient requirements of livestock is able to be determined by inherent genetic variation and environmental factors, mounting interest has been paid to closely regulate the nutrient requirements of domesticated swamp buffalo (henceforth Thai swamp buffalo; Figure 1) to optimize their production in Thailand.

Despite well-recorded studies in maintenance requirement for Thai swamp buffalo, whose performance is similar with the requirement for rumen degradable protein (including non-protein nitrogen, NPN) or protein supplementation either (0.75 g nitrogen (N) or 4.69 g crude protein/ kg BW^0.75^/d [6]) and net energy to convert feed N at energy intake (roughly 2.20 Mcal/kg DM [7,8]), there is no systematic review or investigation on nutrient of Thai growing swamp buffaloes at present. The Thai feeding standard for growing swamp buffalo is currently based on those reports on growing Thai-indigenous beef cattle [9], and early published prescriptions from similar temperate countries observed in Nili-Ravi buffalo [10], Murrah swamp buffalo [11], and general nutrient requirements of beef cattle [12]. Of note, the nutrient requirement of Thai growing swamp buffaloes might be varied compared to respective aforementioned bull species due to differences in genotype, physical growth, growth spurt related to genes, feedstuff quality, topography landscape, and climatic condition [10]. It thus could be a more pronounced nutrient requirement for Thai growing swamp buffaloes; hence, it can be considered an opportune time for addressing this urgent matter in the greater livestock production system and for economic development.

Several sophisticated techniques have been reported to investigate the estimation of nutrient requirements, including energy balance and N mass balance tests with metabolism trials, comparative slaughters on different feeding levels with varied stunning methods, estimation of endogenous N losses in animals calculated by factorial perspective after dietary low- and high-protein diets, and analysis of intake versus performance data in feeding trials [3,8,10,11]. The regression analysis of feeding trial data facilitates the corroborated estimation of the nutrient requirements of producing buffaloes domesticated under prevailing farm feeding conditions [10]. The latter strategy regards a set of statistical processes for estimating the relationships between a dependent variable and one or more independent variables on nutrient requirements in an effective, dynamic manner, in such a way that could generate nutrient requirements at low-cost, without requiring slaughtering of animals; hence, such an approach has been widely used [9,10,11,13,14]. However, the empirical evaluation and prediction of protein requirements for the maintenance and growth of Thai swamp buffaloes has not yet been observed. In the present study, we focussed on protein requirement systems, since this macronutrient is an essential portion of animal feedstuff. Therefore, the study aimed at investigating and predicting protein requirement systems, by providing an abundant energy intake of 2.2 Mcal/kg DM for the maintenance and growth of Thai swamp buffaloes using the comparative prolonged feeding trial for 90 days.

## 2. Materials and Methods

### 2.1. Animal, Diet, and Experimental Design

All procedures involving animals in the present study were approved by Suranaree University of Technology Institutional Animal Care and Use Committee (SUT 4/2558; U1-02632-2559). The experiment was conducted at the Cattle Unit, Suranaree University of Technology farm, Thailand (14°53′21″ N, 102°00′08″ E). Sixteen growing male (bulls) Thai swamp buffaloes, 18–24 months of age and 233 ± 25.0 kg initial weight, were allocated to four experimental groups (*n* = 4) in a randomized complete block design in 2 repeated periods. A period was 45 days and 7 days on the last week, which were set as the time of sample collection. Thai swamp buffaloes were housed individually in metabolic cage (length 3.5 m × width 1.6 m × height 1.9 m) with the ambient temperature ranging from 19.4 ± 2.52 to 29.2 ± 1.39 °C and the average relative humidity was 69.8 ± 6.2%, where each cage had a manual feeder and waterer. Measurement of temperature and humidity was measured twice daily at 07:00 and 20:00 h by using an infrared thermometer (testo 835-H1, Testo SE & Co. KGaA, Bangkok, Thailand). Those circumstances did not tend to buffalo stress. Each buffalo had a properly deworming using ivermectin at 1 mL/30 kg (Vermax, Bangkok, Thailand) on the day before the first adaptation period. Four buffalo each, fed rice straw and concentrate for a 90-day feeding trial (Table 1). Those materials were made freshly per a week as a total mixed ration (TMR) and prepared to avoid protein and energy deficiencies for maintenance and growth following to prior documented requirements [6,7,8,12]. Determination of protein level was based on our previous group result, thereby were 5.4, 6.6, 8.5, and 10.5% crude protein (CP) DM [6]. Provision of net energy was set to each buffalo based on the other buffalo studies [7,8] that were ranged of 2.16–2.25 Mcal/kg DM (1 kg TDN = 3.62 Mcal, Kearl [15]). Buffaloes had free access for clean drinking water throughout the experimental period. Buffaloes were trained-in with a metabolic cage for the 14-day-adaptation period, therefore, buffaloes could adapt the housing set before starting the experiment. All buffaloes were healthy throughout the study.

### 2.2. Sampling

Following the 14-day adaptation period, buffaloes were offered a TMR twice daily at 07:00 and 16:00 h and the daily feed mass was adjusted counted on the dry matter intake (DMI) of the previous day at considering the 90.0% voluntary feed intake. The daily feed intake and refusal diet were recorded daily, and 10.0% of remaining refusal diets were sampled daily (morning and afternoon before next fresh diets offered) and remaining refusal diets were stored at −20 °C for chemical composition.

Buffaloes were weighted monthly, which the first measurement was dedicated as a covariate. On Days 38–44 and 83–89, feces and urine were collected daily by preparing plastic container as given by Paengkoum, et al. [16]. Approximately, 10.0% collected feces and urine each, were collected and acidified to maintain the pH < 3. Feces and urine were composited by each sampling period and frozen at −20 °C for further observation. On Days 45 and 90, 200 mL of rumen fluids were by using a stomach tube connected to a manual pump [17,18,19] 0 and 4 h post-feedings. Monitoring an error by saliva, pump set, and other expectedly barriers during rumen-collecting management were carefully performed. Determination of strict pH, strained through 4 layers of medicinal gauze, and acidified with sulfuric acid had been done before those rumen fluids that were transferred to the laboratory. Each sample was partitioned and stored into three Falcon tubes at −80 °C until analysis of volatile fatty acid (10 mL), N ammonia (5 mL), and rumen microorganism population (20 mL). Furthermore, 10 mL of blood was collected subsequently after rumen fluids collected from the jugular vein into Vacuette tubes (Greiner Bio-One GmbH, Frickenhausen, Germany) containing K3-EDTA (0.47 mol/L). Collected blood samples were centrifuged at 3500 rpm for 20 min at 4 °C (Sorvall Legend XT/XF Centrifuge Series, Thermo Fisher Scientific, Waltham, MA) and plasma was stored at −20 °C to measure blood urea nitrogen (BUN).

### 2.3. Chemical Analysis and Calculation

Samples of diet (TMR), refusal diet, and feces were ground using a mesh size of 1 mm (Retsch SM 100 mill; Retsch Gmbh, Haan, Germany) after dried in an oven at 60 °C for 2 days. Those samples were analyzed for DM, ash, total crude protein (total N × 6.25) following to AOAC [20]. Organic matter (OM) content was calculated as a hundred percentage minus ash percentage, which was achieved after incineration in a muffle furnace at 600 °C for 4 h (AOAC#942.05). Neutral detergent fiber (NDF) and acid detergent fiber (ADF) were detected by the method of Van Soest, et al. [21]. Hemicellulose was calculated as NDF minus ADF. Total digestible nutrient (TDN) and metabolized energy (ME) were estimated using the respective equations: total nutrient digestible = TDN = tdNFC + tdCP + (tdFA × 2.25) + tdNDF – 7 [12]; MP = 1 kg TDN = 3.62 Mcal [15]. In the present study, ingredients of diet, refusal diet, and feces were run (triplicate), and used for chemical analysis and calculation of apparent total tract digestibility.

Determination of total N in feces and urine was measured in triplicate based on prior study [22] with a minor modification [23]. Separated samples of urine were equipped into the high-performance liquid chromatography (HPLC, Hewlett-Packard HPLC system HP series 1100, Agilent technologies, Santa Clara, CA, USA), injected triplicate in C18 reversed phase column with a UV detector wavelength 205 nm, a quat pump, and diode-array detector (G1315A) to determine urinary urine derivatives (PD), creatinine excretion, microbial related to either purine or nitrogen efficiency [24]. The microbial purine based (MPB) was estimated according to Chen, et al. [25] equation, which following factors of digestibility of microbial purines is 0.83, the N content of purine is 70 mg N/mmol, and purine N: total N in mixed rumen microbes is 11.6:100. Thus, microbial N (g N/d) = 70 PB/ (0.116 × 0.83 × 1000) = 0.727PB, where, PB = the corresponding number of microbial purines absorbed (mmol/d). In addition, calculation of daily purine absorption (mmol/d) was determined by performing equation according to Pimpa, et al. [26] equation. Purine derivatives: creatinine (PDC) index was calculated as PDC = PD/creatinine × BW^0.75^.

Collected plasma gathered at 0 and 4 h post-feedings were analyzed for blood urea nitrogen [27]. Similarly, portioned rumen fluids at 0 and 4 h post-feedings were thawing gently. The first portion of rumen fluids was prepared, fixed, and analyzed for volatile fatty acid (VFA) using gas chromatography (Agilent 6890 GC, Agilent Technologies, Wilmington, DE, USA) with a 30 m × 0.25 mm × 0.25 µm column (DB-FFAP) and peak detection was compared and calculated as given by Purba, et al. [28]. Second portion of rumen fluids was centrifuged at 6000 × g for 15 min at 4 °C, and the supernatant was subsequently measured for ruminal ammonia nitrogen (NH_3_-N) using the micro-Kjeldahl methods (Kjeltec 8100, Hillerød, Denmark, AOAC [20]). The last portion of rumen fluids were separated into two aliquots: direct counting cells for bacteria, protozoa, and fungal zoospore using Galyean [29] procedure; roll-tube technique in culture bacteria for groups of cellulolytic, proteolytic, and amylolytic using Hungate [30] method.

### 2.4. Statistical Analysis

All data in this experiment including replication were statistically analyzed as a randomized complete block design (RCBD) following to the general linear model (GLM) procedure of the SAS 9.4 [31]. The model was: Y_ij_= µ + B_i_ + T_j_ + Ɛ_ij_; where; Y_ij_ = the observation, µ = overall mean, B_i_ = effect of block (i: buffaloes), T_j_ = effect of treatment (j: dietary crude protein), and Ɛ_ij_ = random error. The model of covariance of RCBD was: Y_ij_= µ + B_i_ + T_j_ + β (X_ij_ + *x*) + Ɛ_ij_; where Y_ij_ = the observation, X_ij_ = the covariates, response of buffalo in dietary crude protein, µ = overall mean, *x* = mean of covariates B_i_ = effect of block (i: buffaloes), T_j_ = effect of treatment (j: dietary crude protein), β = the regression or slope, adjusted Y by X, and Ɛ_ij_ = random error. The metabolic cage was the experimental unit for each parameter. Despite other parameters, initial weight of buffaloes and number of protozoa at 0 h post- feeding were dedicated as covariate. Comparison of treatment means used Duncan′s new multiple range test and orthogonal contrast analysis which set the method of least significant difference the maximal significant at *p* < 0.05.

## 3. Results and Discussion

The most difficult factor in either investigating or interpreting the protein requirement systems for the maintenance and growth of bovine animals is finding evidence that those requirements are influenced by inherent genetic variation and capability of animals to obtain greater protein synthesis as a considerably energy intake. To the best of the author′s knowledge, the current data show for the first time an investigation that predicts a protein requirement system for the maintenance and growth of 18–24 months old Thai swamp buffaloes. Therefore, the discussion will be based on providing different protein supplies with an abundance of energy supply for the maintenance and growth of Thai swamp buffalo. Protein requirement, nutrient utilization, and microorganism profile are included to elaborate the influences mentioned.

### 3.1. Body Weight and Daily Gain Performance

The different protein supplies with abundance of energy supply fed to Thai swamp buffaloes resulted in improving numbers of final weight and average of daily gain (*p* < 0.05; Table 2). The average of daily gain (ADG) throughout the 90 days of feeding trial was a gradually increased (*p* < 0.05) in buffaloes receiving a greater protein level in diets. Despite buffaloes receiving a level of 5.4% protein in their diet, the present study indicating that those buffaloes had a higher protein efficiency, where buffaloes that received a range of 6.6–10.5% crude protein (CP) were able to get impressive ADG. Our results were in agreement with Chumpawadee, et al. [32] who reported that an increase in ADG of Thai-indigenous yearling heifers was successfully obtained after those heifers were fed diets containing high CP (6.6–13.6%). The assumption that more protein supply is conducive to more weight gain might seem to be true according to the abovementioned. Other bull observations, nevertheless, concluded that excessively dietary CP had no effect on ADG in bull animal performance. For instance, Devant, et al. [33] increased dietary CP level in diets ranging from 14.4–17.3%, and Promkot and Wanapat [34] attempted to decrease the level of protein in diets ranging from 10.5–14.4%, and those observations reported that no gaining weight of crossed heifers occurred. Similarly, Basra, et al. [35] suggested no shift of ADG, when CP level in diets were adjusted from 12.1% to 18.2% of DM in Nili-Ravi buffaloes; thus, suggesting that dietary CP over 10.0% fed to bull animals did not give a positive impact on gaining weight. Dietary CP in the range of 6.6–8.5% constituted an alternative strategy to improve body weight and ADG, without negatively affecting animal performance and farm budgeting. Other discussion had previously reviewed the efficiency of protein-rich diets to contribute to food security, employment, and rural economies [36].

As expected, the aforementioned results of gained weights were derived from a greater feed efficiency (ADG/DMI, CPI or TDNI, *p* < 0.01, Table 2) as a result of receiving more protein supplies in diets. These results were similar with a prior study [37], where Nili-Ravi buffaloes were fed diets containing CP (9.1%) and had a relatively abundant energy intake. These achievements, in turn, lead to lowering the feed efficiency and ADG pattern. If the reduction in a feed intake represents a shift of caloric restriction allowing hyperthermic animals to reduce heat generation [38], a reduction in feed efficiency or ADG pattern was obtained from the limitation of nutrient digestion and metabolism in the rumen host permitting an alleviation of the protein synthesis rate, as indicated from the insufficient energy intake [12].

### 3.2. Nutrient Intake and Nutrient Digestibility

Nutrient intakes beneath requirements by bull animals results in deferred maturity of the reproduction system and slowed down growth rates [9,11,12]. Generally, all of the nutrient intakes on dry-matter (DM) basis increased, when dietary crude protein increased (*p* < 0.01, Table 3). CP intake (g/kg W^0.75^) was a gradually increased in a range of 3.55–9.44 g/kg W^0.75^. It could be noted that no buffalo had restricted the nutrient access per group per experimental design (*n* = 4) in the present study. Thus, the latter outcomes could determine more investigation for nutrient or protein requirement of maintenance and growth in Thai swamp buffaloes. Moreover, the rate of nutrient intakes could be influenced by several factors such as rumen capacity, ruminal metabolic level (factual VFAs), digestion rate, physiological animal, and nutrient requirement setting [39]. We know of few serial observations regarding dietary DM intake influenced by dietary CP as similar as the present study. Comparative studies observed in Nili-Ravi buffaloes [35,37], Thai-indigenous heifers [32,40], and Murrah buffaloes [7,11] showed that either protein or nitrogen intake was varied when those bull animals were fed to the extent of the dietary CP. In addition, the rumen capacity, including limited passage rate, suggested that the microbial turnover was increasing and leading to a reduced efficiency of microbial protein [12]. In this case, low feed intake might have occurred that led to animals′ severe nutrient deficiency.

Nutrient digestibility, especially the digestibility of protein, is essential—for both bacterial crude protein (BCP) and undegradable intake protein (UIP) during absorbing the metabolizable protein [12]. In the present study, digestibility rates of DM, OM, total digestible nutrient (TDN), neutral detergent fiber (NDF) were lower, whereas digestibility of protein was higher when dietary crude protein increased (*p* < 0.05, Table 3). Although, the present results were in contrast with previous study [11] that Murrah buffaloes showed the greater digestibility rates of DM, OM, and protein. Diets were consisted of berseem, wheat straw, and concentrate mixture without inclusion of urea. Previously, the inclusion of urea from 10 to 30 g/kg in the diet composition led to increase the digestibility rates both of OM and protein [41]. To note, the present study provided dietary CP increases with simultaneously increasing urea levels in the range of 3.7–8.0 g/kg DM in the buffalo diets (Table 1). This suggests that the presence of urea in bull animal diets might enhance deamination and modulate the apparent of protein; however, the exceeding urea supplementation of the offered diets might have had considerably impact on ureagenesis. A generous hepatic ureagenesis becomes indispensable for ruminants to refrain poisoning from absorbed ammonia [42]. Furthermore, considering carbohydrate digestion (TDN) in the rumen could be the most accurate predictor of BCP synthesis [12]. The present study expected to provide the protein and carbohydrate digestions, so that the provision of 6.6% protein, the inclusion of 5.5 g/kg urea and the availability of energy intake at 2.2 Mcal/kg DM in the diet (Table 1) suggested the optimum level for buffaloes to obtain greater digestibility rates of protein and TDN.

### 3.3. Ruminal Fermentation and Blood Urea Nitrogen

Ruminal fermentation end-product (pH, NH_3_-N, VFAs) and blood metabolite such as blood urea nitrogen (BUN) are crucial parameters to assess whether animals are meeting their nutrient requirement without reducing the animal responses [7,11,35,37,43]. Increasing the protein content in Thai swamp buffalo diets enriched the concentration of NH_3_-N in the rumen (*p* < 0.05), but surprisingly did not affect the ruminal pH (Table 4). The ruminal pH of Thai swamp buffaloes fed the diets in the present study was unvaried, as the provision of different protein contents and the pH rate values (6.7–7.1) were expected. The current result was similar to prior observations [32,40]. To note, inclusion of urea in diets of the present study suggested a similar finding by Wanapat, et al. [44] who added urea to a buffalo diet in a range of 15–30 g/kg concentrate. The ruminal pH was in the normal range from 6.8 to 6.9, which provided the optimal rumen circumstance for the rumen host to grow and to ferment nutrient digestion, especially protein. Hence, it might be that the rumen-buffering capacity did not occurred, but that even the ruminal NH_3_-N changed. Further studies are needed to better explain the regression between rumen-buffering capacity, pH, and ruminal NH_3_-N on manipulating protein supply in buffalo diets.

Furthermore, BUN concentration was shifted by dietary CP (*p* < 0.01; Table 4). As aforementioned a shift of ruminal NH_3_-N, the mean of ruminal NH_3_-N was at normal level (11.3–22.2 mg/dL) and considered safe for rumen host to avoid microbial turnover increasing [45]. However, another study suggested that a shift of ruminal NH_3_-N related to altered BUN concentration as modulating dietary CP for bull animals [40,46,47]. Here, the mean of BUN in the present study was varied (*p* < 0.01, 13.7–26.8 mg/dL) and those numbers were still considerably safe for ruminal bull animals [40,47]. Moreover, there was high relationship between nitrogen intake as varied protein intake, ruminal NH_3_-N, and BUN (Figure 2). Ruminal NH_3_-N concentration increases when protein intake increases that relates to faster protein degradation than synthesis, higher dietary rumen-degradable protein (RDP) [47], or an imbalance of fermentable energy [41,47,48]. This occur suggesting that ammonia would be accumulated to rumen fluid and result in exceedingly concentration. As aforementioned, all of the ammonia including urea retention absorbed in lumen of the gut was removed by the liver with, as a result, a net splanchnic flux of zero to perform detoxification of ammonia by the liver [42], and finally excreted via urine in to high levels of urine nitrogen [40,41].

Total VFA in the present study was varied in the range of 79.0–85.0 mM, and the VFA fraction including acetate and propionate was changed but the butyrate remained unchanged, where samples of rumen fluids observed at 0 and 4 h post-feedings by dietary CP (Table 4). In general, the influences of the dietary CP occurred in the buffaloes fed the diet that consisted of 10.5% protein at 4 h post-feeding (*p* < 0.05). A shift of VFAs relates to the ruminal host on digesting and metabolizing the nutrient source in diet, here, buffaloes could manifest their efficiency of feed utilization that is represented by increasing a number of VFAs. A higher VFAs might be due to a greater digestibility by VFA-producing bacteria. More CP intake at 10.5% showed the highest protein digestibility compared with other protein supplementations (Table 3). However, those effects are time dependent. Since fermentable energy availability (TDN, g/kg W^0.75^) was obtained in a similar pattern, the present results reflect that the dietary CP allowed the VFA-producing bacteria to rapidly accelerate the protein breakdown on nutrient digestion and the latter fermentable products were forwarded to hydrolysis and stored at a shift of VFAs as a main source of buffalo energy [44,47]. Additionally, there was a shift in VFAs produced in the rumen toward more propionate with corresponding to reduce in acetate and remain in butyrate (Table 4). The acetate portion decreased when the propionate increased, indicating that the dietary CP in the present study altered the fermentable carbohydrate to be a major substrate for acetate fraction; however, a higher propionate suggested that propionate-producing bacteria was dominant to synthesize fermentable carbohydrate more propionate in gluconeogenesis by the pentose phosphate pathway produces a nicotinamide adenine dinucleotide phosphate [18,49,50]. These occurrences might relate to the reduction of NDF digestion (Table 3). The observed NDF digestion playing a role in the VFA-producing substrate is line with data by Vorlaphim, et al. [51], who reported no changes for propionate portion in rumen fluids at 0 h, 2 h, and 4 h after feedings when NDF intake (g/d) was similar. It, thus, can be suggested that this declining availability of fermentable carbohydrate might relate to decrease acetate portion as the first major VFA absorbed from the rumen to have somewhat distinctive metabolic shift [28,49,50]. This achievement was similar with the previous study [32] and was expected in the present study because of the altered carbohydrate digestion.

### 3.4. Nitrogen Balance, Purine Derivative Excretion, Microbial-Related N Characteristic, and Rumen Microorganim Population

The provision of increased dietary CP in Thai swamp buffaloes increased the size of feces (*p* < 0.05, Table 5). This occurrence was expected due to the efficiency of dietary CP which was suggested, below 10% DM CP in the diet, as aforementioned [32,33,34,35]. Hence, the present results could be corroborating previous reports. This possibly occurred due to a higher dietary CP; the increasing fecal size might also be due to the likely enlarged DM intake and slightly dwindled number of DM digestibility (Table 3), so as to excrete lowered undigested dietary CP formed in a higher volume of feces. Therefore, a significant difference in fecal N among diets in the present study could be attributed to the endogenous losses from digestive tracts; however, it might not be varied among bull animals [15].

Concurrently, the current scenario resulted in N contents determined in N intake, N excretion, N absorption, and N retention as indicated a higher N amount of urinary and fecal (*p* < 0.01). It has been well-documented that increasing protein content in diet inevitably increases N intake and this surpasses the numbers of N retention and N excretion in Thai-indigenous heifers [9,40], Murrah buffaloes [11], and Nili-Ravi buffaloes [46]. Notably, Thai swamp buffaloes fed to increased dietary CP as a depiction in the present study showing that there is relationship between N intake, N retention, and both of N absorption and excretion (g/kg W^0.75^; Figure 3). These recent findings interpreting that N retention was ranged from 0.07 to 0.45 g N/kg W^0.75^ and N absorption ranged from 0.21 to 0.96 g N/kg W^0.75^, when the Thai swamp buffaloes consumed the increased dietary CP levels. This relationship gave evidence that the provision of increased dietary CP among bull animals resulted in an increase in the N amount in feces, nitrogen fecal, and urine, if the fermentable energy availability was insufficient supply.

It has been discussed that N excretion via urine is increased with an increasing protein level in diet (N intake), which may relate to the RDP ration in dietary CP [41,52]. To note, high RDP ratios in dietary CP have been showing due to the increase of N in urine, but this occur led to a decrease N balance [46]. The utilized nitrogen parameter such as nitrogen or protein degradability has a main effect on urinary N output because of excess soluble N in the rumen from diets with high RDP [41]. Several factors, namely N origin, degree of N intake, degree of energy intake, age, sex, and physiological animal related to metabolic perspective could be considered as profoundly influencer on shifting nitrogen balance [53]. Nitrogen balance defines the animal’s status to meet their protein requirement, as metabolized protein excess (supply minus requirements; NRC [12]). However, excessively N retaining no longer gives a metabolic benefit due to those N absorbed from the lumen of the gut into the portal vein, which must be partially removed by the liver to avoid a large hepatic ureagenesis as discussed above [42].

The urinary purine derivative (PD) excretion (mmol/d or µmol/kg W^0.75^) increased when dietary CP in Thai swamp buffaloes increased (*p* < 0.05; Table 6). Concentration of PD parameters (allantoin, uric acid, xanthine, and hypoxanthine) in the present study was in the range of 16.3–31.3 mmol/d and it was expected to be a similar range as reported in other buffalo investigations [26,54]. In addition, there was a significant increase (*p* < 0.05) in allantoin and total PD, and non-significant responses or remaining unchanged in relatively uric acid, xanthine and hypoxanthine excretions that observed in urine as influence of increased dietary CP. These achievements in the present study were similar to previous studies [26,41], which sizes of allantoin and uric acid excretions were relatively normal for bull animals as influenced protein content in diets. Moreover, the present study by adjusting dietary CP seemed to have no influence on urinary excretion of creatinine (µmol/kg W^0.75^). However, dietary CP shifted ratio of A:C, PD:C and PDC index (*p* < 0.01; Table 6). Creatinine excretion might be influenced by the amount of protein intake by animals and the present study showed increased pattern of DM intake as well as concomitantly protein content in diets increased (Table 1 and Table 3). However, the recent findings were not in line with previous studies that were observed in Zebu cattle and swamp buffaloes [55], and Murrah buffaloes [56]. To compare, those bull animals had been to show the difference of creatinine excretions due to the marginal variations among studies. Variation of animals, diets, experimental managements were expected to differ the characteristic and profile in creatinine shift [55]. We speculated also that breed or species specific and more closely correlated with muscle mass than body weight might differ the creatinine size, and not depend on dietary intake. Thus, the present study could be assumed to reconfirm that the dietary intake of animals did not attribute to vary the creatinine excretion, if those fed to similar in diet concern and group of selected breeds.

To determine and predict the protein requirement for the maintenance and growth of Thai swamp buffaloes on any given feeding program, it is necessary to establish the characteristic of microbial nitrogen supply and microbial nitrogen efficiency. Adjusting dietary CP was expected to increase linearly microbial purine base (MPB) flow (mmol/d), microbial N supply (g N/d), and microbial N efficiency (g N/kg nutrient utilization) in urinary Thai swamp buffaloes (*p* < 0.01; Table 6). However, a solely microbial N efficiency in terms of CP intake remained unchanged. Our results were in agreement with the previous study of Paengkoum et al. [41]; their group assessed that microbial N synthesis grew up by adding urea from 10 to 30 g/kg steam-treated oil palm fronds. As mentioned above, urea stimulated to increase protein availability in rumen. Kim et al. [47] reported rumen richer in fermentable protein derived from dietary CP (9.2 vs. 11.2%DM) and solely RDP fraction (52.3 vs. 79.8%CP) resulting in plentiful microbial N supply to omasum. It may indicate microbial N turnover decreasing due to a greater N source. Exceedingly high CP supplementation at 13.5 to 19.4% DM had been reported to show an inclination of omasal flow of total non-NH_3_-N bacteria from 425 to 480 g/d [57]. More efficiency gains had been achieved by only increasing RDP level (10.6–13.2%) in the diets tending to enrich total non-NH_3_-N bacteria [58]. It could be assessed that impressive numbers of total non-NH_3_-N bacteria flowing from rumen to omasum due to dietary CP increased as protein intake and its degradability increased, but this efficiency of microbial protein synthesis had seemed to depend on the age of rumen itself. The age-related succession of rumen microbial communities, especially *Bacteroidetes* and *Proteobacteria* populations, defined the shift of ruminant animal productivity as growing higher in presence of older age rumen [59]. In other words, animal age might have influence on differing microbial efficiency. Moreover, the present study had provided the experimental design in numerous levels of CP and considerably energy supply. This was prepared because we still assumed that any protein catabolism converting to amino acid seemed to have sufficient fermentable energy availability. The traditional meta-analysis regarding microbial efficiency in rumen had been previously discussed to relate protein synthesis is based on ruminal carbohydrate as a main energy source in rumen [13]. The high-energy and low-protein diets shifted the available N for microbial growth in limited microbial protein synthesis [60]. Of note, therefore, several factors such as the availability of carbohydrates and N in rumen, ruminal pH, physiological effects, sources and levels of N components, and other stabilizing ruminal fermentations had been discussed to have a substantial role on modifying the efficiency of microbial protein in rumen [13,60,61].

Despite similar pH in rumen, dietary CP increased relatively fungal zoospore and total bacteria in rumen of Thai swamp buffaloes (*p* < 0.05; Table 7). Cellulolytic bacterial population had quadratic and cubic effects (*p* < 0.05). Amylolytic bacterial population had a linear effect (*p* < 0.05). However, increasing CP content in diets did not affect proteolytic bacterial population in the rumen of Thai swamp buffaloes neither observed in 0 and 4 h post-feedings. The inclination of increased numbers in those recently mentioned findings were similar patterns as a well-documented in those prior reports [62,63]. There may be a big question mark in which the proteolytic bacterial population did not get affection by adjusting dietary CP, while the digestibility of CP (Table 3) was a greater appearance. The possibly reason might relate to the protein metabolism itself in rumen as present objective delivered. Protein metabolism in the rumen is metabolically manifested by activity of ruminal microorganisms to have a corresponding effect towards the nutrient utilization. Here, protein utilization is depending on the structure of nutrient and it is defining whether having a susceptibility to microbial proteases and, thus, its degradability. Bach, et al. [64] had pointed out several findings by reviewing the nitrogen metabolism in the rumen. Ruminal protein degradation is shifted by pH and the major microbial population. Proteolytic population represents the protein degrader in rumen and its activity changes as pH changes with high-forage dairy cattle-type rations, but not in high-concentrate beef-type rations. It, thus, indicating that proteolytic population degrades the protein substrates in post-feeding and finally shifts to the amino acid in rumen. However, some amino acid, such as Ile, Leu, and Phe which are synthesized by rumen microorganisms tend to limit the protein degradation in rumen [9,53,64]. The second reason pointing to a lack of determining proteolytic population was in a recent alternative and complementary technique of the present study, where the measurement of proteolytic counting used the roll-tube technique observed in rumen culture that is no longer accurate. It was recently suggested that bacterial cells, including N distribution therein, could be affected by indigenously factor in rumen such as rate of fermentation [64]. As consequence, further studies are needed to greater explain the protein utilization by proteolytic population in rumen determined by using the sequence technique to define extracellular prokaryotic diversity in bovine rumen [65].

### 3.5. Nitrogen or Crude Protein Requirement

The net nitrogen or crude protein requirements for maintenance and growth of Thai swamp buffaloes were determined based on ADG (g ADG/kg W^0.75^) and N intake (g N/kg W^0.75^; Figure 4). It depicted the linear regression between dietary nitrogen or crude protein and the net requirements for maintenance and growth of Thai swamp buffaloes, as equation = 0.0725ADG + 0.8663 (R^2^ = 0.577, *p* < 0.001, *n* = 16). Nitrogen requirement could be estimated, the nitrogen intake in which ADG (equally at 0) was nitrogen requirement for maintenance, ADG index (at slope) was nitrogen requirement for growth of Thai swamp buffaloes. Of note, it could be mathematically calculated following to 0.866 g N/kg W^0.75^ or equivalent to 5.41 g CP/kg W^0.75^/d (maintenance requirement), and 0.073 g N/g ADG or equivalent to 0.46 g CP/g ADG/d (growth). The recently findings regard of the net CP requirement for growth of Thai swamp buffaloes was higher than it reported in NRC [12], but the maintenance was lower. Besides, the present results were slightly higher for maintenance of buffaloes and lower to roughly at 24.0% for growth of buffaloes compared to those reported in Kearl′s report (maintenance, (5.24 g CP/kg W^0.75^; growth, 0.65 g CP/g ADG), as recommendation CP requirements of ruminants in developing countries [15]. It is possible that these differences were at the UIP presence of the applied diet as a deciding factor to have influence on the unaffected nutrient apparent tract digestibility and the population of ammonia-producer bacteria. Rumen fermentation alleviated with the greater number of UIP in diet, but UIP intensified ammonia retention as higher efficiency of nitrogen utilization in rumen. For instance, Paengkoum et al. [9] had estimated the metabolized protein (MP) requirement to increase one g/kg BW^0.75^ that was 0.34 g MP/kg BW^0.75^ of Thai-indigenous growing cattle and MP requirement for maintenance was 2.77 g/kg BW^0.75^. It was suggested to supplement not over 10.0% CP DM in diet with proportional ratio of UIP and degradable intake protein (DIP, 35:65), which resulted in the optimum growth performance for growing Thai-indigenous beef cattle. Several previous studies also had been testing to prove the net nitrogen requirement for maintenance and growth among bull animals having a age-related succession of rumen microbial communities [59], breed and sex [35,37], and balanced content of protein and energy [7,56]. Collectively, the present findings could corroborate other possibly factors, protein requirement of bull animals might relate to specific domestication and climatic condition as Thai swamp buffaloes investigated in the present study.

## 4. Conclusions

In conclusion, the net crude protein requirement for maintenance of Thai swamp buffaloes was lower compared to NRC 2001. The net crude protein requirement for growth of Thai swamp buffaloes was higher compared to NRC 2001. The present results suggested that crude protein requirement increases when body weight increases. An increased dietary crude protein resulted in an increased number at nutrient intake, nutrient digestibility, total volatile fatty acid, blood urine nitrogen, ammonia absorption, urinary purine derivative, and microbial related ammonia. Adjusting dietary crude protein in the range of 6.6–8.5% crude protein on dry-matter basis in diet is suggested as an effective strategy to develop body weight and average daily gain in Thai swamp buffaloes. More studies that use miscellaneous ages, larger sample size, inherent genetic variation, and varied production systems should be further carried out, so as to shed light on the changes in those dietary requirements of crude protein either in maintenance or growth stages.

## Figures and Tables

**Figure 1 animals-11-01405-f001:**
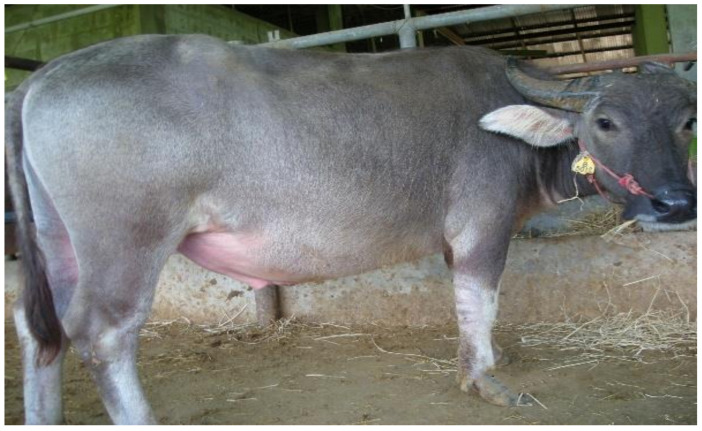
Thai swamp buffalo.

**Figure 2 animals-11-01405-f002:**
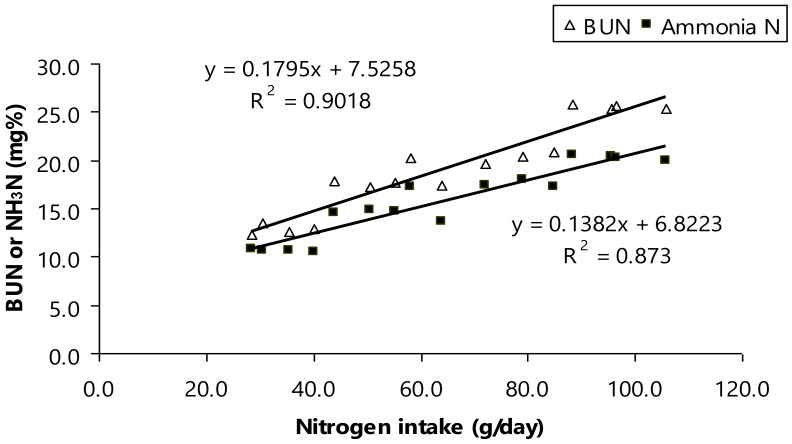
Relationship between nitrogen intake and either blood urea nitrogen (BUN) or ruminal ammonia nitrogen (NH_3_-N).

**Figure 3 animals-11-01405-f003:**
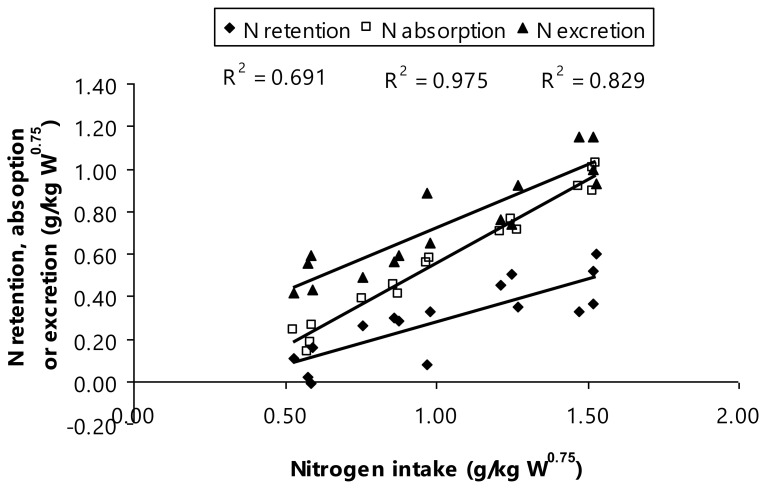
Relationship between nitrogen intake and nitrogen utilization.

**Figure 4 animals-11-01405-f004:**
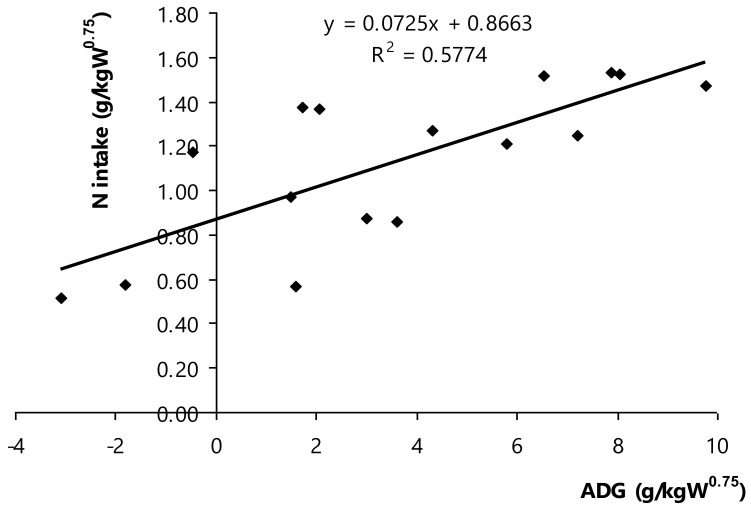
Relationship between average daily gain (ADG) and N intake in Thai swamp buffaloes.

**Table 1 animals-11-01405-t001:** Ingredient and chemical composition of experimental diets (g/kg dry matter, DM).

Item	Crude Protein Level			
	5.4%	6.6%	8.5%	10.5%
Ingredient, g/kg DM				
Rice straw	619.0	631.0	667.9	655.0
Cassava pulp	148.8	153.6	140.2	123.8
Ground corn	227.6	178.1	110.2	108.1
Soybean meal	0	31.0	73.8	104.4
Urea	3.7	5.5	7.1	7.9
Premix ^1^	0.9	0.8	0.8	0.8
Chemical composition				
Dry matter, g/kg as basis	873.7	874.5	874.9	876.5
Organic matter, g/kg DM	893.5	892.4	884.6	882.3
Crude protein, g/kg DM	54.2	66.4	85.4	104.5
Neutral detergent fiber, g/kg DM	569.4	551.6	577.0	570.2
Acid detergent fiber, g/kg DM	367.3	361.5	384.3	385.6
Hemicellulose, g/kg DM	202.1	190.1	192.6	184.6
Total digestible nutrient, g/kg DM ^2^	624.0	622.0	601.5	598.2
Metabolizable energy, Mcal/kg DM ^3^	2.2	2.2	2.2	2.2

^1^ The premix contained (per 5 kg): vitamin A, 20 mIU; vitamin D3, 2 mIU; vitamin E, 20 IU; Mn, 80 g; Zn, 50 g; Fe, 120 g; Cu, 10 g; Se, 0.25 g; Co, 1 g; I, 2.5 g. ^2^ Calculated from National Research Council (NRC) [12]; TDN = tdNFC + tdCP + (tdFA × 2.25) + tdNDF − 7. ^3^ Metabolizable energy; calculated from Kearl [15]: 1 kg total digestive nutrient (TDN) = 3.62 Mcal.

**Table 2 animals-11-01405-t002:** Effect of dietary crude protein on body weight and daily gain performance in Thai swamp buffaloes.

Item	Crude Protein level				SEM	Significance	
	5.4%	6.6%	8.5%	10.5%		Linear	Quadratic
INW, kg	233.8	234.0	234.0	232.8	6.74	0.962	0.914
FW, kg	229.3 ^c^	248.3 ^bc^	260.5 ^ab^	279.0 ^a^	8.50	0.002	0.977
AW, kg	231.5 ^b^	241.1 ^ab^	247.3 ^ab^	255.9 ^a^	7.34	0.039	0.947
ADG, kg/d	−0.05 ^c^	0.16 ^b^	0.29 ^b^	0.51 ^a^	0.05	0.0001	0.912
ADG/DMI, g/kg	−14.20 ^c^	31.89 ^b^	54.49 ^b^	92.07 ^a^	9.91	0.0001	0.678
ADG/CPI, g/g	−0.28 ^b^	0.48 ^a^	0.62 ^a^	0.86 ^a^	0.17	0.001	0.149
ADG/TDNI, g/kg	−22.49 ^c^	51.14 ^b^	90.65 ^b^	153.73 ^a^	16.36	0.0001	0.755

INW: initial weight; FW: final weight; AW: average weight; ADG: average daily gain; DMI: dry matter intake; CPI: crude protein intake; TDNI: total digestible nutrient intake; SEM: standard error of mean; ^a–c^ values on the same row under each main effect with different superscript differ significantly (*p* < 0.05).

**Table 3 animals-11-01405-t003:** Effect of dietary crude protein on nutrient intake and apparent digestibility in Thai swamp buffaloes.

Item	Crude Protein Level				SEM	Significance	
	5.4%	6.6%	8.5%	10.5%		Liner	Quadratic
Dry matter intake							
kg/d	4.17 ^c^	4.97 ^b^	5.25 ^ab^	5.60 ^a^	0.16	0.0001	0.190
g/kg W^0.75^	70.43 ^b^	81.25 ^a^	84.13 ^a^	87.67 ^a^	1.94	0.0001	0.094
% BW	1.8 ^b^	2.1 ^a^	2.1 ^a^	2.2 ^a^	0.05	0.0005	0.118
Crude protein intake							
kg/d	0.21 ^d^	0.33 ^c^	0.46 ^b^	0.60 ^a^	0.01	0.0001	0.482
g/kg W^0.75^	3.55 ^d^	5.42 ^c^	7.34 ^b^	9.44 ^a^	0.21	0.0001	0.608
TDN intake							
kg/d	2.57 ^b^	3.12 ^a^	3.15 ^a^	3.35 ^a^	0.09	0.0003	0.095
g/kg W^0.75^	43.57 ^b^	50.93 ^a^	50.49 ^a^	52.51 ^a^	1.31	0.001	0.072
Other nutrient intakes							
OM, kg/d	3.70 ^c^	4.44 ^b^	4.65 ^ab^	4.94 ^a^	0.14	0.0001	0.140
NDF, kg/d	2.45 ^c^	2.77 ^bc^	3.06 ^ab^	3.21 ^a^	0.11	0.0005	0.421
ADF, kg/d	1.65 ^c^	1.82 ^bc^	2.07 ^ab^	2.17 ^a^	0.08	0.0007	0.708
Hemicellulose, kg/d	0.80 ^b^	0.96 ^a^	1.00 ^a^	1.03 ^a^	0.04	0.001	0.121
Apparent digestibility							
Dry matter, %	60.8 ^ab^	63.2 ^a^	57.5 ^ab^	56.4 ^b^	1.77	0.044	0.359
Organic matter, %	66.6 ^ab^	69.7 ^a^	65.1 ^b^	63.0 ^b^	1.26	0.026	0.073
Crude protein, %	36.4 ^c^	52.5 ^b^	58.2 ^ab^	63.3 ^a^	2.76	0.0001	0.082
TDN, %	91.1 ^a^	91.9 ^a^	89.4 ^b^	88.6 ^b^	0.47	0.001	0.124
NDF, %	53.4 ^ab^	60.0 ^a^	47.9 ^b^	44.8 ^b^	3.19	0.029	0.170
ADF, %	51.4	50.4	45.9	44.0	2.67	0.055	0.870
Hemicellulose, %	57.5 ^ab^	68.0 ^a^	51.8 ^b^	46.9 ^b^	3.91	0.025	0.085

BW: body weight; TDN: total digestible nutrient; OM: organic matter; NDF: neutral detergent fiber; ADF: acid detergent fiber; SEM: standard error of mean; ^a–d^ values on the same row under each main effect with different superscript differ significantly (*p* < 0.05).

**Table 4 animals-11-01405-t004:** Effect of dietary crude protein on ruminal fermentation end-product and blood metabolite in Thai swamp buffaloes.

Item	Crude Protein Level				SEM	Significance	
	5.4%	6.6%	8.5%	10.5%		Linear	Quadratic
Ruminal pH							
0 h post-feeding	7.08	7.05	7.08	6.98	0.04	0.152	0.365
4 h post-feeding	6.88	6.75	6.70	6.75	0.07	0.224	0.259
NH_3_-N concentration, mg/dL							
0 h post-feeding	10.39 ^d^	12.44 ^c^	16.14 ^b^	18.44 ^a^	0.26	0.0001	0.636
4 h post-feeding	10.80 ^d^	16.37 ^c^	18.88 ^b^	22.12 ^a^	0.27	0.0001	0.002
BUN concentration, mg/dL							
0 h post-feeding	11.94 ^d^	16.02 ^c^	19.34 ^b^	24.34 ^a^	0.30	0.0001	0.207
4 h post-feeding	13.69 ^d^	19.02 ^c^	21.34 ^b^	26.83 ^a^	0.21	0.0001	0.706
Total VFA, Mm							
0 h post-feeding	82.55	88.34	86.84	85.43	3.95	0.695	0.386
4 h post-feeding	79.01 ^b^	81.56 ^b^	81.56 ^b^	85.00 ^a^	0.86	0.001	0.615
VFA profile, mol/100 mol							
Acetic acid							
0 h post-feeding	68.80	69.09	69.63	68.75	0.56	0.876	0.319
4 h post-feeding	69.14 ^a^	68.43 ^a^	68.69 ^a^	65.96 ^b^	0.69	0.015	0.182
Propionic acid							
0 h post-feeding	19.62	19.35	19.30	20.36	0.55	0.406	0.258
4 h post-feeding	18.78 ^b^	20.04 ^b^	19.95 ^b^	22.46 ^a^	0.61	0.003	0.331
Butyric acid							
0 h post-feeding	11.57	11.56	11.07	10.89	0.39	0.186	0.845
4 h post-feeding	12.07	11.53	11.36	11.57	0.34	0.294	0.294

NH_3_-N: ammonia; BUN: blood urea nitrogen; VFA: volatile fatty acid; SEM: standard error of mean; ^a–d^ values on the same row under each main effect with different superscript differ significantly (*p* < 0.05).

**Table 5 animals-11-01405-t005:** Effect of dietary crude protein on excreta volume and nitrogen balance in Thai swamp buffaloes.

Item	Crude Protein Level				SEM	Significance	
	5.4%	6.6%	8.5%	10.5%		Linear	Quadratic
Excreta volume							
Urine, L/d	3.55	3.62	4.48	4.80	0.57	0.103	0.828
Feces, kg DM/d	1.67 ^b^	1.83 ^b^	2.23 ^a^	2.44 ^a^	0.12	0.001	0.839
Urine nitrogen							
g/d	8.08 ^c^	10.06 ^c^	20.93 ^b^	32.55 ^a^	2.17	0.0001	0.054
g/kgW^0.75^	0.14 ^c^	0.16 ^c^	0.34 ^b^	0.51 ^a^	0.36	0.0001	0.073
Feces nitrogen							
g/d	21.79 ^c^	25.19 ^bc^	30.77 ^ab^	35.17 ^a^	2.12	0.001	0.819
g/kgW^0.75^	0.36 ^c^	0.41 ^bc^	0.49 ^ab^	0.55 ^a^	0.03	0.0009	0.873
Nitrogen intake							
g/d	33.65 ^d^	53.28 ^c^	73.41 ^b^	96.50 ^a^	2.36	0.0001	0.482
g/kgW^0.75^	0.57 ^d^	0.87 ^c^	1.17 ^b^	1.51 ^a^	0.03	0.0001	0.603
Nitrogen excretion							
g/d	29.87 ^c^	35.24 ^c^	51.70 ^b^	67.73 ^a^	2.85	0.0001	0.095
g/kgW^0.75^	0.50 ^c^	0.57 ^c^	0.83 ^b^	1.06 ^a^	0.04	0.0001	0.110
Nitrogen absorption							
g/d	11.86 ^d^	28.10 ^c^	42.64 ^b^	61.33 ^a^	2.33	0.0001	0.610
g/kgW^0.75^	0.21 ^d^	0.46 ^c^	0.68 ^b^	0.96 ^a^	0.04	0.0001	0.714
Nitrogen retention							
g/d	3.78 ^b^	18.03 ^a^	21.70 ^a^	28.78 ^a^	3.56	0.0008	0.339
g/kgW^0.75^	0.07 ^b^	0.29 ^a^	0.35 ^a^	0.45 ^a^	0.05	0.001	0.330

SEM: standard error of mean; ^a–d^ values on the same row under each main effect with different superscript differ significantly (*p* < 0.05).

**Table 6 animals-11-01405-t006:** Effect of dietary crude protein on purine derivative and microbial-related N characteristic in Thai swamp buffaloes.

Item	Crude Protein Level				SEM	Significance	
	5.4%	6.6%	8.5%	10.5%		Linear	Quadratic
Purine derivative (PD)							
Allantoin (A)							
mmol/d	10.52 ^b^	14.83 ^b^	13.77 ^b^	25.26 ^a^	1.90	0.002	0.092
µmol/kg W^0.75^	180 ^b^	240 ^b^	225 ^b^	405 ^a^	39.8	0.005	0.163
%	65.3 ^b^	77.1 ^a^	78.3 ^a^	80.6 ^a^	2.54	0.003	0.096
Uric acid							
mmol/d	3.00	2.09	1.50	3.82	0.75	0.588	0.059
µmol/kg W^0.75^	56.25	34.00	24.50	61.50	15.8	0.931	0.093
%	11.3	10.8	8.5	12.1	2.81	0.183	0.107
Hypoxanthine							
mmol/d	0.97	0.83	0.65	0.50	0.16	0.456	0.076
µmol/kg W^0.75^	18.00	10.75	8.25	13.00	3.54	0.298	0.125
%	5.7 ^a^	3.5 ^b^	2.9 ^b^	2.7 ^b^	0.62	0.008	0.143
Xanthine							
mmol/d	1.80	1.61	1.74	1.40	0.22	0.297	0.737
µmol/kg W^0.75^	3.00	27.50	27.50	25.00	2.88	0.275	1.00
%	11.7 ^a^	8.6 ^ab^	10.3 ^a^	4.6 ^b^	1.58	0.023	0.432
Total PD							
mmol/d	16.28 ^b^	19.18 ^b^	17.51 ^b^	31.32 ^a^	2.59	0.004	0.064
µmol/kg W^0.75^	287 ^c^	312 ^b^	287 ^b^	503 ^a^	55.4	0.034	0.121
Creatinine (C)							
mmol/d	30.08	25.63	19.68	32.56	4.23	0.971	0.062
µmol/kg W^0.75^	552	416	322	519	86.7	0.628	0.087
Ratio A:C	0.39 ^b^	0.59 ^ab^	0.71 ^a^	0.78 ^a^	0.06	0.001	0.339
Ratio PD:C	0.58 ^b^	0.76 ^ab^	0.90 ^a^	0.97 ^a^	0.07	0.003	0.455
PDC index	34.86 ^b^	47.34 ^ab^	55.97 ^a^	61.42 ^a^	4.98	0.003	0.498
Microbial related N							
MPB flow, mmol/d	37.99 ^c^	56.44 ^b^	42.99 ^b^	155.26 ^a^	24.6	0.013	0.088
Microbial N supply, g N/d	27.60 ^b^	41.03 ^b^	31.25 ^b^	112.87 ^a^	17.9	0.012	0.086
Microbial N efficiency							
g N/kg DOMR	17.70 ^b^	20.38 ^b^	15.90 ^b^	54.23 ^a^	8.34	0.020	0.061
g N/kg OMI	8.07 ^b^	9.23 ^b^	6.57 ^b^	22.39 ^a^	3.72	0.038	0.080
g N/kg DMI	7.19 ^b^	8.26 ^b^	5.81 ^b^	19.76 ^a^	3.29	0.040	0.082
g N/kg CPI	0.14	0.12	0.06	0.19	0.04	0.769	0.154
g N/kg TDNI	36.04 ^c^	48.92 ^bc^	60.50 ^b^	85.26 ^a^	5.24	0.0001	0.286

PD: purine derivative; MPB: microbial purine base; DOMR: digestibility of organic matter fermented in rumen; OMI: organic matter intake; DMI: dry matter intake; CPI: crude protein intake; TDNI: total digestive nutrient intake; SEM: standard error of mean; ^a–d^ values on the same row under each main effect with different superscript differ significantly (*p* < 0.05).

**Table 7 animals-11-01405-t007:** Effect of dietary crude protein on microbial group in rumen of Thai swamp buffaloes.

Item	Crude Protein Level				SEM	Significance	
	5.4%	6.6%	8.5%	10.5%		Linear	Quadratic
Direct count							
Protozoa, ×10^5^ cell/mL							
0 h post-feeding	7.81	4.63	6.56	4.81	2.10	0.471	0.740
4 h post-feeding	9.25	4.75	8.63	7.38	3.17	0.904	0.620
Fungal zoospores, ×10^7^ cell/mL							
0 h post-feeding	1.72 ^b^	3.35 ^ab^	4.14 ^ab^	6.54 ^a^	0.97	0.006	0.699
4 h post-feeding	3.36	3.63	4.51	3.35	0.56	0.834	0.244
Total bacteria, ×10^9^ cell/mL							
0 h post-feeding	1.33	1.40	1.39	1.80	0.18	0.106	0.338
4 h post-feeding	1.01 ^b^	1.09 ^b^	1.23 ^b^	1.69 ^a^	0.13	0.012	0.111
Roll-tube technique							
Amylolytic bacteria, ×10^6^ CFU/mL							
0 h post-feeding	2.06 ^b^	12.81 ^ab^	6.94 ^b^	39.25 ^a^	9.61	0.036	0.291
4 h post-feeding	6.19	4.19	19.88	18.44	6.91	0.124	0.968
Cellulolytic bacteria, ×10^8^ CFU/mL							
0 h post-feeding	0.98 ^b^	1.20 ^b^	3.15 ^a^	1.27 ^b^	0.53	0.266	0.079
4 h post-feeding	0.68 ^b^	0.90 ^ab^	1.75 ^a^	0.73 ^b^	0.27	0.431	0.047
Proteolytic bacteria, ×10^6^ CFU/mL							
0 h post-feeding	3.81	3.19	2.69	5.92	1.83	0.496	0.320
4 h post-feeding	3.38	3.19	4.13	5.00	1.20	0.306	0.668

SEM: standard error of mean; ^a–d^ values on the same row under each main effect with different superscript differ significantly (*p* < 0.05). CFU: colony forming unit.

## Data Availability

The data presented in this study are available on request from the corresponding author.

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
