# Peer review of "Empirical Evaluation and Prediction of Protein Requirements for Maintenance and Growth of 18–24 Months Old Thai Swamp Buffaloes"

_animals, 2021, doi:10.3390/ani11051405_

Round 1
Reviewer 1 Report
The authors highly improved their paper which is now acceptable in present form.
Author Response
Dear Reviewer #1
Thank you for your cooperation.
Regards,
Corresponding author

Reviewer 2 Report
I read the work with poor enthusiasm and I find it not sufficient structured from the point of view of the experimental model (very few animals respect to the attended results).
The scientific production relating to the protein and energy requirements of the different types of buffaloes (at different class of age) in the world is very large and the one reported in the text seems too focused on the Thai swamp buffalo.
The simple summary is too simple but still manages to give a sufficient overview of the content of the paper.
The abstract is well done and provides a clear overview of the paper’s content.
In lines 26-28 authors cite: “Interpretation of increased gain in the bovine animals is difficult to be investigated due to inherent genetic variation to meet their requirement for energy and protein, and those may relate to the bull species, e.g., Thai swamp buffalo.”
Since this statement is not true worldwide, in my opinion the text should be changed to: "In some geographical areas and in certain breeding situations, the interpretation of increased gain in the bovine is difficult to be investigated. The inherent genetic variations to meet their energy and protein needs vary as a function of inherent genetic differences, making it difficult to accurately assess in bull species, e.g., Thai swamp buffalo.”
INTRODUCTION
The introduction describes incompletely the state of the art relating buffalo's knowledges around the world about protein requirements for maintenance and growth of buffalo 18-24 months old.
The metabolism of water buffalo (Mediterranean) has been studied for many years and the results are very detailed for growing heifers in large farms and for specialized milk production. No bibliographical notes are reported for these.
for example:
- metabolic status and ovarian function in buffalo heifers fed a low energy or high energy;
- Protein nutrition and nitrogen balance in buffalo cows.
- Nitrogen and phosphorus utilization and excretion in dairy buffalo intensive breedin
MATERIALS and METHODS
The experimental design is not convincing. Four animals per group are few and the age range is too large (6 months) to express an average weight of 233 kg with a standard deviation of only 25 kg. Please convince me on these points.
All percentage values must have one or two decimal places and the chosen criterion must be used in the full text.
In line 107 the authors cite: "… the ambient temperature ranging from 106 19.4 ± 2.52 to 29.2 ± 1.39 °C and the average relative humidity was 69.75 ± 6.24%,..."
Are the temperature values the minimum and maximum values of the morning and of the night? Do they express an average value of the entire experimental period? In any case, I do not believe that the humidity does not vary with thermal excursions of 10 °C.
Line 115 The crude protein values reported in the text do not correspond to those in table 1.
Edit Tables to be consistent with the text.
RESULTS
The results are well structured and explained in detail.
CONCLUSIONS
On line 577 add an "s" to swam.
Author Response
Dear Reviewer #2
Authors have performed the extensive revision as your suggestions line by line, please see the attachment.
Thank you for your cooperation.
Regards,
Corresponding author

This manuscript is a resubmission of an earlier submission. The following is a list of the peer review reports and author responses from that submission.
Round 1
Reviewer 1 Report
The paper is of good quality and originality, I only recommend some suggestions in the discussion, which can be seen in the paper that I send you.

Author Response
Thank you for giving us the opportunity to submit a revised draft of my manuscript titled “Empirical evaluation and determination of protein requirements for maintenance and growth of 18–24-month-old Thai swamp buffaloes” to Veterinary Sciences. We appreciate the time and effort that reviewer 1 have dedicated to providing your valuable feedback on our manuscript. We are grateful to the reviewer 1 for their insightful comments on our paper. We have been able to incorporate changes to reflect most of either questions or suggestions provided by the reviewer 1.
Comments and/or suggestions:
L86: And the relation degradeble carbohydrates and degradable protein and Microbial protein syntesis
Response: Thank you. However, present study was discussed about center providing different protein supplies with an abundance of energy supply for maintenance and growth of Thai swamp buffalo. Protein requirements, nutrient use, and microorganism profile are described to analyze the mentioned influences. Therefore, we are afraid to add suggested sentence due to it falling out of context in the present study.
L95: 38 days to adaptation???
Response: Sorry for this misunderstanding. However, it is clearly that we state an adaptation period for 14 days (Please see L114). A day of 38 days was assigned as feeding trial and remaining 7 days was for collecting samples.
L99: have a proof of this (stress)
Response: Thank you. We recorded that all buffaloes had abundance of eating behavior during adaptation period and of course all buffaloes were healthy throughout the study (please see L112).
Table 1: and level of undegradable protein and degradable protein an carbohydrates??? because change level of cassava, urea and corn.
Response: Thank you for your note. You might be right that difference in level of cassava, urea, and corn may have effect on undegradable protein and degradable protein. However, in the present study, organic protein content was set to increase following to the treatments.
Table 1: Diet with 10.71% have more rice straw and less Acid detergent fiber????
Response: Thank you for the note. Table 1 has been revised.
Table 2: why buffaloes have weight loss is only protein or relationship with energy or type of protein ??? And have less ADG/DMI , ADG/CPI, ADG/TNDI?????
Response: We specified that a reduction in feed efficiency or ADG can be obtained from the limitation of nutrient digestion and metabolism in the host’s rumen host and reducing the protein synthesis. Also, we add a sentence with a reference to confirm that reducing weight loss might be related to cardiometabolic effect (Please see L376-380).
L267: Problaprobably due to the relationship between degradable protein and degradable carbohydrates and microbial protein synthesis.
Response: We have included your suggestion (Please see L422-428).
Somewhere in page 8: with more urea you need more fast-degrading carbohydrates.
Response: We have included your suggestion (Please see L437).
Table 4: Due to the excessive content of soluble and degradable nitrogen and an inadequate relation with carbohydrates of fast and medium degradation
Response: Thank you for the suggestion. We have included your suggestion (Please see L435-437).
Table 5 (volume): probably you didnt had differences in
Response: You might be right, difference in nitrogen losses in urine and feces is provided in the next response.
Table 5 (urine nitrogen row): Greater losses of Nitrogen in urine and feces to greater consumption of nitrogen, in urine it is due to endogens losses, increase in the concentration of NH3 and conversion to urea with greater use of energy and this is due to an excess of soluble nitrogen and an inadequate relationship with fast-degrading carbohydrates, reducing microbial protein synthesis
Response: We have included your suggestion (Please see L538-544).
Reviewer 2 Report
In my opinion the paper need only very few corrections an/or additions as indicated in the attached file

Author Response
Thank you for giving us the opportunity to submit a revised draft of my manuscript titled “Empirical evaluation and determination of protein requirements for maintenance and growth of 18–24-month-old Thai swamp buffaloes” to Veterinary Sciences. We appreciate the time and effort that reviewer 2 have dedicated to providing your valuable feedback on our manuscript. We are grateful to the reviewer 2 for their insightful comments on our paper. We have been able to incorporate changes to reflect most of either questions or suggestions provided by the reviewer 2.
Comments and/or suggestions:
L63: please, after " ..... reproduction and growht" add this reference: Campanile G., Di Palo R., Infascelli F., Gasparrini B, Neglia G., Zicarelli F., D’Occio M.J. (2003) Influence of rumen protein degradability on productive and reproductive performance in buffalo cows. Reproduction Nutrition Development, 43, 557-556.
Response: Thank you. We have added the reference (Please see L64)
Table 1: please report the Chemical Composition as g/kg DM instead of % DM.
Response: We have revised Table 1 as suggestion.
References: please add the reference mentioned above
Response: We have added the initial reference appropriately (L720-721).
Reviewer 3 Report
The manuscript is confusing and difficult to understand. The introduction must be implemented, material and methods reviwieved as well as table. Several mistakes are in the table 1. Results should be rewritten to make them more understanding.
Author Response
Thank you for giving us the opportunity to submit a revised draft of my manuscript titled “Empirical evaluation and determination of protein requirements for maintenance and growth of 18–24-month-old Thai swamp buffaloes” to Veterinary Sciences. We appreciate the time and effort that reviewer 3 have dedicated to providing your valuable feedback on our manuscript. We are grateful to the reviewer 3 for their insightful comments on our paper. We have been able to incorporate changes to reflect most of either questions or suggestions provided by the reviewer 3.
Comments and/or suggestions:
The manuscript is confusing and difficult to understand. The introduction must be implemented, material and methods reviwieved as well as table. Several mistakes are in the table 1. Results should be rewritten to make them more understanding.
Response: Thank you for the suggestion. In table 1, you are right, we have revised it appropriately. The text to fix the grammatical errors and improve the overall readability of the text before we send it for revision have been done by using MDPI English language editing (certificate attached).
